# C-UNet: Complement UNet for Remote Sensing Road Extraction

**DOI:** 10.3390/s21062153

**Published:** 2021-03-19

**Authors:** Yuewu Hou, Zhaoying Liu, Ting Zhang, Yujian Li

**Affiliations:** 1Faculty of Information Technology, Beijing University of Technology, Beijing 100124, China; houyuewu@foxmail.com (Y.H.); zhaoying.liu@bjut.edu.cn (Z.L.); liyujian@bjut.edu.cn (Y.L.); 2School of Artificial Intelligence, Guilin University of Electronic Technology, Guilin 541004, China

**Keywords:** UNet, complementary UNet, fixed threshold, dilated convolution, remote sensing

## Abstract

Roads are important mode of transportation, which are very convenient for people’s daily work and life. However, it is challenging to accuratly extract road information from a high-resolution remote sensing image. This paper presents a road extraction method for remote sensing images with a complement UNet (C-UNet). C-UNet contains four modules. Firstly, the standard UNet is used to roughly extract road information from remote sensing images, getting the first segmentation result; secondly, a fixed threshold is utilized to erase partial extracted information; thirdly, a multi-scale dense dilated convolution UNet (MD-UNet) is introduced to discover the complement road areas in the erased masks, obtaining the second segmentation result; and, finally, we fuse the extraction results of the first and the third modules, getting the final segmentation results. Experimental results on the Massachusetts Road dataset indicate that our C-UNet gets the higher results than the state-of-the-art methods, demonstrating its effectiveness.

## 1. Introduction

Road, as a vital special feature in remote sensing images, includes highways, urban-rural roads, byway, and so on. Road extraction has important significance in many fields, such as automatic road navigation, disaster relief, urban planning, and geographic information update [1]. However, it is a challenging task because of the noise, occlusions, and complexity of the strcture of roads in remote sensing image [2].

There are mainly two types of images, i.e., aerial infrared thermography and remote sensing images, used to road segmentation. The aerial infrared thermography can be monitored 24 h a day, without being affected by strong light. However, the contrast of infrared images is low, lacking of image details. This disadvantages make it difficult to extract road segmentation [3]. Remote sensing images are less limited by ground conditions, real time transmission, and its detection range is large. All these advantages make it more suitable to extract roads.

Scholars have studied the road extraction of remote sensing image and put forward a variety of methods. These methods can be roughly divided into four types: data-based method, graph cut method, semi-automatic method, and automatic method [4]. Data-based method generally extracts roads from remote sensing images with the information of data. For example, Wegner et al. [5] suggested to get the road segmentation results with conditional random fields. They significantly improved both the per-pixel accuracy and the topological correctness of the extracted roads on two different datasets. Maurya et al. [6] proposed a clustering method to do road segmentation. The method extracts roads very rapidly and give satisfactory results with small number of simple images. Mattyus et al. [7] utilized Markov random fields to finish road segmentation. They demonstrated their approach outperforming the state-of-the-art in the two datasets they collected. These methods have certain limitations, such as poor generalization ability for different types of roads, and cannot handle multi-scale roads.

Graph cut method belongs to the unsupervised learning. It relies on the color features to extract roads information. For example, Cheng et al. [8] proposed a graph cut based probability propagation approach to extract road from complex remote sensing images. They achieved better performance both in qualitative and quantitative comparisons in the two datasets they collected. Cheng et al. [9] introduced a graph cut method with multiple features. They got better performance than other methods on 25 images. Yuan et al. [10] presented a novel graph cut method and obtained higher results than other methods. Although these methods alleviate the traditional data-based problems to a certain extent, they cannot achieve better results for images with multiple colors on the road [4].

The semi-automatic road extraction, man-machine interaction is used in the process of road feature extraction and recognition [11]. The main idea is as follows: firstly, the initial seed point of the road is set manually, and initial direction is set if necessary; then judgment and recognition is conducted by the computer according to the corresponding rules, and at the same time man-machine interaction is appropriately used to ensure the accuracy of recognition. Commonly-used methods include dynamic programming [12,13], models based on snakes [14,15] or active contour [16,17], models based on active testing [18], template matching [19,20], etc. Constant manual intervention is needed in the semi-automatic road extraction, increasing the workload of remote sensing image interpreters [21]. Additionally, artificial auxiliary information is required in the formation and repair of road segments and in the continuous stage of road segments [22]. The semi-automatic road extraction objectively improves accuracy rate, while reducing the work efficiency. Therefore, it is not conducive to promotion.

As for automatic road feature extraction method, roads are automatically interpreted and recognized by extracting and understanding road image features [23]. Specifically, the features of the roads in the image are firstly analyzed, and then the roads are automatically identified by pattern recognition methods [24]. Among them, convolutional neural networks (CNNs) based methods are the most representative [25,26,27,28]. Zhong et al. [29] proposed a CNN model that combines low-level fine-grained features and high-level semantic features to extract road and building targets in satellite images. Alshehhi et al. [30] proposed a patch-based CNN model for extracting road and building parts simultaneously from remote sensing imagery. Subsequently, a road extraction method based on the fully convolutional network (FCN) model appeared. Varia et al. [31] applied a deep learning technique FCN-32 for extracting road parts from extremely high-resolution Unmanned Aerial Vehicle (UAV) imagery. Kestur et al. [32] presented U-shaped FCN based on the FCN to extract roads from UAV images. Panboonyuen et al. [33] presented a technique based on landscape metrics and the exponential linear unit function to extract road objects from remote sensing imagery. Hong et al. [34] applied a block based on richer convolutional features for road segmentation from high-resolution remote sensing imagery. Cheng et al. [35] proposed the cascaded end-to-end CNN model based extracting road centerlines from remote sensing imagery. CNN-based models can automatically explore road characteristics by using strong generalization ability, the arbitrary function fitting ability and high stability, then predict the probability value of pixel-level road images through discriminant function [36,37]. They achieved better performance than the other three types of methods. However, they rely heavily on abundant images, and the number of remote sensing images is generally limited.

To segment roads with limited remote sensing images, Ronneberger et al. proposed the UNet based on FCN by deepening the number of network layers and adding cross layer connections between corresponding layers [38]. UNet obtained the highest mean cross-over ratio and the smallest warping error, respectively, in the International Symposium on Biomedical Imaging (ISBI) Cell Tracking Challenge [39] and Electron Microscopy (EM) Segmentation Challenge [40] in 2014 and 2015, respectively.

At present, most of the mainstream remote sensing image road extraction models are based on UNet. For example, by adding residual module [41] to the original UNet network, Zhengxin et al. got a deep ResUNet which obtained the highest recall rate at the Massachusetts dataset [42]. Furthermore, Oktay et al. added the Attention Gate to the decoder of UNet, and proposed an attention UNet model, which highlighted the segmented targets by suppressing the characteristic responses of unrelated background regions without introducing many model parameters [37]. Zhou et al. proposed the DinkNet34 model [43]. Based on the UNet model, it expanded the receptive field while maintaining the resolution by simultaneously using dilated convolution module [44], and won the championship of the DeepGlobe 2018 Challenge [42].

All of the above models conduct remote sensing image road extraction by means of a single network. However, a single network limits the performance of road network extraction because it cannot handle the roads of various shapes, lengths and widths. Therefore, a method of remote sensing image road extraction based on complement UNet (C-UNet) was proposed in this paper. The model has two characteristics: one the one hand, it sets a fixed threshold to erase the pixels in the first segmentation result; on the other hand, it introduces the multi-scale dilated convolution UNet (MD-UNet) to extract more difficult road information.

There are mainly two differences between UNet and C-UNet. Firstly, UNet segments the roads from remote sensing images with a single network, without considering the diversity of road width and length, while C-UNet utilized two UNet variations, i.e., UNet and MD-UNet, to successively segment the roads. The former is used to extract easier road information, and the latter is forbidden to extract complement and more difficult road information. Secondly, UNet cannot obtain larger receptive fields, while C-UNet armed with the dilated convolution operation to obtain larger receptive fields, making it more suitable for high-resolution remote sensing images.

The main contributions of the study were summarized as follows:(1)To improve the accuracy of remote sensing image road extraction, we propose a complement UNet model, called C-UNet, for high-resolution remote sensing image road extraction. The model used standard UNet and MD-UNet to extract road information in remote sensing image successively, then fused the results of segmentations, and, lastly, obtained the final segmentation result, which was better than the state-of-the-art methods.(2)A kind of erasing method for fixed significant area was proposed. By using a fixed threshold, it erased part of the road area in the remote sensing image extracted by standard UNet, so that the network could extract finer and weaker road area for the second time.(3)By comparing our model with the UNet SERIES models proposed in recent years, the experimental results showed our model achieved better results than the previous state-of-the-art models, verifying the effectiveness of our model. In addition, some ablation studies were established to verify the overall structure and major modules.

The major contents of the rest part of the study are as follows: UNet is briefly introduced in Section 2. The model is introduced in detail in Section 3. The experimental results are shown in Section 4. Discussion is showed in Section 5. Summary and conclusion are given in Section 6.

## 2. UNet

UNet is an improved fully convolutional network model, and its structure is similar to shape U [38]. The detailed architecture of UNet is shown in Figure 1. Compared with other convolutional neural networks, UNet requires less training sets and has higher segmentation accuracy. As seen from the Figure 1, it is composed of encoder and decoder, which are symmetrical with the symmetry axis of the intermediate layer. The encoder extracts image features through convolutional layers down-sampling (also known as pooling) layers. By comparison, the decoder conducts up-sampling of feature images, and there are cross-layer connections between the corresponding encoder and decoder layers, which can help the up-sampling layer to recover the details of the image.

Specifically, the image feature information that the encoder extracts through convolutional layer is composed of 3×3 convolutional layer, ReLU function and 2×2 max-pooling layer. Four times of down-sampling are conducted. The size of the feature images decreases and the number of channels doubles after each pooling operation. The decoder performs up-sampling by 2×2 deconvolution layer (or transposing convolution) and gradually recovers the image information. Corresponding to the encoder part, the decoder part completes up-sampling for four times. The size of the feature images increases and the number of channels reduces by half after each up-sampling. The detailed location information that can be more effectively saved with shallow network assists segmentation through concatenation of the corresponding feature pattern of encoder and decoder. The UNet contains a total of 23 convolutional layers.

## 3. C-UNet

In this section, we first introduce the overall architecture of C-UNet. Then, we describe the erasing module, multi-scale dilated convolution UNet, and the fusion module, in turn. Finally, the loss function that trained our model was given.

### 3.1. Overall Network Architecture

To impove the performance of road extraction from high-resolution remote sensing images, we propose C-UNet with four modules. First of all, remote sensing images were input into standard UNet for road extraction in the first module and the first segmentation results were obtained. Secondly, a fixed threshold value was set to erase the pixels that exceeded the threshold value and the segmentation result after erasing was obtained. Then, the segmentation results after erasing were input into the multi-scale dilated convolution UNet for road segmentation in the third module. Lastly, the segmentation results of the first module and the third module were fused, obtaining the final segmentation results. As for the advantage, the model completed road segmentation sequentially through standard UNet and multi-scale dilated convolution UNet. The former was used to extract road information that was simpler to be segmented, while the latter was used to extract the road information that was not extracted by the former, namely the road information that was more difficult to be segmented. The flowchart and overall architecture of C-UNet are shown in Figure 2 and Figure 3, respectively.

To be specific, let $X^{H\times W\times C}$ denote a remote sensing image and $F_{unet}$ denote the output by the standard UNet, the process could be expressed with the following equation:(1)$$F_{unet}=UNet\left(X^{H\times W\times C}\right).$$

The first segmentation result ${\mathrm{Pre}}_{1}$ could be obtained by putting $F_{unet}$ into the sigmoid function and applied a binary operation, and it could be expressed in the following form:(2)$${\mathrm{Pre}}_{unet}=\sigma \left(F_{unet}\right),$$
(3)$${\mathrm{Pre}}_{1}=binarized\left({\mathrm{Pre}}_{unet}\right),$$
where σ(·) denotes the sigmoid function, and binarized(·) is the binary operation.

Secondly, a fixed threshold was set to erase the pixels in the feature pattern $F_{unet}$ that were larger than the threshold δ, and the erased feature pattern was obtained, which could be expressed in the following way:
(4)$$F_{unet}^{{}^{\prime }}=E\left(F_{unet},\delta \right),$$
where E(·) stands for the erasing operation.

Later on, the erased feature pattern was input into the MD-UNet, and the second segmentation result was obtained. The process could be expressed as with the following equation:(5)$${\mathrm{Pre}}_{md-unet}=F_{md-unet}\left(F_{unet}^{{}^{\prime }}\right),$$
(6)$${\mathrm{Pre}}_{2}=binarized\left({\mathrm{Pre}}_{md-unet}\right),$$
where ${\mathrm{Pre}}_{2}$ stands for the segmentation results of the third module, and binarized(·) denotes the binary operation.

Finally, the segmentation results of the first module and the third module were fused and the fused result ${\mathrm{Pre}}_{final}$ was obtained, as the final segmentation results. It could be expressed as follows:(7)$${\mathrm{Pre}}_{final}=Fusion\left({\mathrm{Pre}}_{1},{\mathrm{Pre}}_{2}\right).$$

### 3.2. Erasing Methods

The partial road areas that had already been segmented by the standard UNet were erased by using threshold erasing method, so that the model, namely multi-scale dilated convolution UNet in the third module could segment the road areas that were difficult to be segmented.

Specifically, let δ denote the threshold value, $F_{unet}\left(i,j\right)$ represent the value of the row *i* and column *j* of the output feature images of the standard UNet, $F_{i,j}^{{}^{\prime }}$ indicate the value of the row *i* and column *j* of the feature images after erasing. The threshold erasing could be expressed as follows:(8)$$F_{unet}^{{}^{\prime }}=\left\{\begin{array}{cc} F_{unet}\left(i,j\right), & \mathrm{if}\;F_{unet}\left(i,j\right)\leq \delta \\ 0, & \mathrm{if}\;F_{unet}\left(i,j\right)>\delta \end{array}\right..$$

The use of threshold erasing not only erased wrong segmentation areas but also made MD-UNet segment the road area that was difficult to be segmented.

### 3.3. Multi-Scale Dilated Convolution UNet

Dilated convolution is frequently applied in semantic segmentation [45,46], target detection, and other fields [47,48]. It can enlarge the receptive field and capture multi-scale context information. Considering the roads of remote sensing image have different shapes, width and length, multi-scale information is quite crucial in remote sensing image network extraction. The dilation rate, as one parameter of dilated convolution, refers to the number of dilation filled in the standard convolution kernel. Dilated convolution, by means of the parameter, expands the receptive field without introducing additional parameters. In order to access to more abundant multi-scale features, the multi-scale dilated convolution module with different dilation rates was used, with the specific structure shown in Figure 4. As a multi-scale dilated convolution module, dilated block can enlarge the receptive field of feature images and obtain more detailed local information. The architecture of the multi-scale dilated convolution UNet is shown in Figure 5.

### 3.4. Fusion Process

The road information extracted in the first module by standard UNet was relatively easy to extract, while road information extracted in the third module by multi-scale dilated convolution UNet was relatively thin and weak. The results of the two modules were complementary to each other. In order to obtain complete and accurate segmentation results, it was necessary to fuse the two segmentation results after obtaining the results of UNet and multi-scale dilated convolution UNet. The fusion process is shown in the following formula.
(9)$$Fusion\left({\mathrm{Pre}}_{1},{\mathrm{Pre}}_{2}\right)=\left\{\begin{array}{cc} 0, & \mathrm{if}\;Pre_{1}\left(i,j\right)=Pre_{2}\left(i,j\right)=0\\ 1, & \;\;otherwise \end{array}\right.,$$
where ${\mathrm{Pre}}_{1}$ and ${\mathrm{Pre}}_{2}$ represent the two segmentation images to be fused, and $Pre_{1}\left(i,j\right)$ and $Pre_{2}\left(i,j\right)$ represent the pixel values of the two feature images to be fused at the position (i,j).

As seen from the above formula, after complementary fusing of the unsegmented regions in the two segmentation results of the first and the third module, we could achieve more complete and accurate road information, further improving the segmentation performance.

### 3.5. Loss Function

Binary cross-entropy was used as the target function in the study. Let ${\mathrm{Pre}}_{final}^{p}$ represent the *p*-th image predicted by the model, ${\mathrm{GT}}^{p}$ denote the ground truth of the image *p*, $gt_{i,j}^{p}$ be the pixel values of ground truth at position (i,j), $pre_{i,j}^{p}$ indicate the pixel values of the image predicted by the model at the position (i,j), *N* express the number of training samples, and *W* and *H* suggest the width and height of the image, respectively. Then, the binary cross-entropy loss could be expressed in the following form:(10)$$\begin{array}{cc} L_{bce} & =BCELoss({\mathrm{Pre}}_{final}^{p},{\mathrm{GT}}_{final}^{p})\\ \; & =-\sum_{p=1}^{N}\sum_{i=1}^{W}\sum_{j=1}^{H}\left[\left(gt_{i,j}^{p}\times \log pre_{i,j}^{p}\right)+\left(1-gt_{i,j}^{p}\right)\times \log\left(1-pre_{i,j}^{p}\right)\right] \end{array}.$$

## 4. Experimental Results

In this section, we first introduce the public dataset used in the experiment, Massachusetts Road dataset [42]. Then, we give the implementation details and evaluation indexes. Next, we perform the ablation studies to verify the effectiveness of the model and its submodules. Finally, we compare our model with the state-of-the-art models to prove the superiority of our method. All the experiments were realized with the Pytorch (version of 1.3.0) framework and were conducted on Nvidia Tesla K40c GPU server, with the memory size of 11 GB, the CPU of Intel Xeon E5-2643, and the operating system of Windows 7.

### 4.1. Dataset

Massachusetts Road dataset is usually used for road extraction of remote sensing images [42]. It contains 1171 images, with the resolution of 1500×1500, and binary segmentation label (black represents non-road area, and white represents road area). It covers a wide range of areas, involving more than 2600 km${}^{2}$ of urban, suburban and rural areas in the United States. Figure 6 displayes the Geo-referenced Map of Massachusetts.

Due to the limited video memory of the server used in the experiment, remote sensing images with the resolution of 1500×1500 could not be directly used for training. Thus, the images in the dataset were pretreated by dividing each image with the resolution of 1500×1500 and its corresponding ground truth image into 9 with the resolution of 512×512. The specific dividation steps were as follows: first, a 512×512 bounding box template was taken and slid on the 1500×1500 image (base image); the images on the four corners and the most middle were cut out; the images in the middle of each adjacent two templates in the four bounding boxes on the four corners were taken, in turn. The specific dividing method is shown in Figure 7. The regions of the 9 images overlapped each other. Furthermore, based on analysis, it was found that the ground truth of some images was wrongly marked, so such images were excluded. In the end, 8960 images with the resolution of 512×512 were obtained. According to the division of the training set, verification set and test set of the original data set, 8361 training images, 126 verification images, and 433 test images were obtained, respectively. Figure 8 shows an example of the image in the Massachusetts road dataset and its corresponding ground truth.

### 4.2. Implementation Details and Evaluation Indicators

#### 4.2.1. Implementation Details

We use mini-batch stochastic gradient descent to optimize the parameters of our model. During training, the parameters of our model were optimized through Adam method in all experiments of the study. The number of training iterations was 15, and the initial learning rate was set as $2\times 10^{-4}$. After 8 training, the learning rate became $2\times 10^{-5}$. The size of the input image was 512×512, and the size of the mini-batch was set as 1.

#### 4.2.2. Evaluation Indicators

Two evaluation indexes, namely the Mean Intersection over Union (mIOU) [49] and Mean Dice coefficient (mDC) [50], were used in the experiment to assist the evaluation of the quality of different models.

mIOU refers to the overlap rate of the generated candidate boxes and the original marker boxes, that is, the ratio between the intersection and the union. A greater mIOU means a better segmentation result. Let $p_{ii}$ denote the number of correct elements predicted, $p_{ij}$ represent the number of ones with the true values of *i* and the predicted value of *j*, $p_{ji}$ be the number of ones with the true values of *j* and the predicted value of *i*, and *k* mean the number of categories to be classified. Then, the mIOU could be expressed as follows:(11)$$mIOU=\frac{1}{k+1}\sum_{i=0}^{k}\frac{p_{ii}}{\sum_{j=0}^{k}p_{ij}+\sum_{j=0}^{k}p_{ji}-p_{ii}}.$$

As a measurement function of set similarity, the Dice coefficient can be used to calculate the similarity between the segmentation images and ground truth. A larger mDC stands for a better segmentation result. Let ${\mathrm{Pre}}_{final}^{p}$ represent the segmentation result of image *p*, ${\mathrm{GT}}_{final}^{p}$ denote the ground truth of image *p*, and *N* represent the number of training samples. Then, the mDC could be expressed in the following form:(12)$$\begin{array}{cc} mDC & =Dice({\mathrm{Pre}}_{final}^{p},{\mathrm{GT}}_{final}^{p})\\ \; & =\frac{\sum_{p=1}^{N}\left|{\mathrm{Pre}}_{final}^{p}\cap {\mathrm{GT}}^{p}\right|}{\sum_{p=1}^{N}\left(\right|{\mathrm{Pre}}_{final}^{p}\left|+\right|{\mathrm{GT}}^{p}\left|\right)} \end{array}.$$

### 4.3. Ablation Study

In this subsection, we first explore the effect of different erasing methods and erasing thresholds on network performance. Then, we verify the effectiveness of multi-scale dilated convolution UNet and the fusion process. Finally, we compare our model with the previous state-of-the-art models to demonstrate the superiority of our model.

#### 4.3.1. Ablation Study on the Method of Erasing

Firstly, we discuss the influence and the necessary the erasing method on the performance of C-UNet. We use two erasing methods after the first module of UNet to obtain the first segmentation result. The first one was threshold erasing, represented as C-UNet-threshold (i.e., C-UNet). In this method, we set the threshold with 0.7 in advance, and then erase the pixels in the segmentation result that greater the threshold to get the segmentation results after erasing. The other one was random bounding box block erasing [51], expressed as C-UNet-random. In this method, we use a rectangular box with random size and all pixels of 0 to randomly block a certain region in the segmentation result, so as to achieve the purpose of erasing. Besides, we omit the erasing method after the first module and represented the model as C-UNet_no_erase. The specific experimental results of these three methods are shown in Table 1, Figure 9 and Figure 10.

From Table 1, Figure 9 and Figure 10, we can find that:(1)The values of mIOU and mDC obtained by C-UNet-random were 0.613 and 0.739, while the values of mIOU and mDC obtained by C-UNet-threshold were 0.635 and 0.758, respectively. Comparing to the result of C-UNet-random, the results of C-UNet-threshold were improved by 0.021 and 0.017, respectively. It is possibly because the more obvious segmentation regions in the segmentation results of the first module were erased by threshold erasing method, making the UNet in the third module, i.e., the multi-scale dilated convolution UNet, pay more attention to those targeted regions that were difficult to be segmented.(2)The rectangular box random erasing method used different rectangular boxes to randomly erase the segmentation results in the first module. At this time, the erasing area layout was not targeted, directly making the UNet segmentation in the third module be purposeless. Therefore, fixed erasing could help improve the segmentation performance of C-UNet.(3)The values of mIOU and mDC obtained by C-UNet were, respectively, 0.635 and 0.758, which were improved by 0.006 to the result of C-UNet_no_erase (0.629 and 0.752 for mIOU and mDC, respectively). It was indicated that better segmentation results were obtained in C-UNet with erasing process than C-UNet without erasing process, i.e., the erasing process was necessary to improve the performance of C-UNet.

#### 4.3.2. Ablation Study on the Threshold of Erasing

Secondly, we explore the influence of different erasing thresholds on the segmentation performance of C-UNet. The erasing thresholds of 0.5, 0.7, and 0.9 were selected, in turn, and the corresponding segmentation results of C-UNet were obtained. The specific experimental results are shown in Table 2, Figure 11 and Figure 12.

Based on Table 2, Figure 11 and Figure 12, under the thresholds of 0.5, 0.7, and 0.9, the mIOU obtained by C-UNet were 0.632, 0.635, and 0.636, respectively, and the mDC obtained by C-UNet were 0.755, 0.758, and 0.756, respectively. With the threshold of 0.7, the mIOU and mDC obtained by C-UNet were 0.635 and 0.758, respectively, higher than the results when the threshold was 0.5. Relatively to the results under threshold of 0.9, mIOU decreased by 0.001, while mDC increased by 0.02. Therefore, the threshold of 0.7 was finally selected as the erasing threshold in the study after comprehensive consideration of the results.

#### 4.3.3. Ablation Study on Dilated UNet

Later on, we argue about the effectiveness of MD-UNet. For the selection of UNet in the third module, UNet [38], Non-local Block [52], FCN [29], and MD-UNet [43] were used for experiments. Therefore, we got four different models represented as UNet_UNet, UNet_Non-local, UNet_FCN, and UNet_MD-UNet (i.e., C-UNet), respectively. Table 3, Figure 13 and Figure 14 show their results.

As shown in Table 3, Figure 13 and Figure 14, the values of mIOU obtained by UNet_UNet, UNet_Non-local, UNet_FCN, and UNet_MD-UNet(C-UNet) was 0.622, 0.615, 0.606, and 0.635, respectively, and the results of mDC obtained by them was 0.744, 0.739, 0.730, and 0.758, in turn. UNet_MD-UNet(C-UNet) obtained the highest mIOU and mDC, respectively, possibly because multi-scale dilated convolution could fuse feature images from different scales and obtain more detailed segmentation results in the decoder.

#### 4.3.4. Ablation Study on the Fusion

Finally, we illustrate the effectiveness of fusion module of C-UNet. We compare the segmentation results of the first module (i.e., UNet) of C-UNet, the third module (i.e., MD-UNet) of C-UNet, and the segmentation results after fusing the results of the two modules. Table 4, Figure 15 and Figure 16 show the corresponding results.

As seen from Table 4, Figure 15 and Figure 16, the values of mIOU and mDC corresponding to the UNet in the first module were 0.614 and 0.738, respectively, and those corresponding to the MD-UNet in the third module were 0.618 and 0.743, respectively. Besides, finally, the fusion module got mIOU and mDC of 0.635 and 0.758, 2.1% higher than that of UNet and 1.7% higher than that of MD-UNet. This indicates that it is necessary to fuse the results of two different modules.

### 4.4. Comparison of C-UNet with Other Models in Remote Sensing Image Road Extraction

In this subsection, we verify the effectiveness of the C-UNet method by comparing it with existing models. The state-of-the-art models in the previous remote sensing road segmentation task, namely UNet [38], ResUNet [36], AttUNet [37], and DinkNet34 [43] were selected for comparison. The corresponding results are shown in Table 5, and the corresponding segmentation examples are listed in Figure 17 and Figure 18.

As seen from Table 5 and Figure 17 and Figure 18, the values of mIOU obtained by UNet [38], ResUNet [36], AttUNet [37], DinkNet34 [43], and C-UNet were 0.599, 0.600, 0.616, 0.607, and 0.635, respectively, and the corresponding value of mDC was 0.725, 0.721, 0.740, 0.733, and 0.758, respectively. C-UNet obtained the highest mIOU and mDC, respectively. Compared with the results of the other 4 models, the mIOU obtained by C-UNet was improved by 0.036, 0.035, 0.019, and 0.028, respectively, and mDC was improved by 0.033, 0.037, 0.018, and 0.025, respectively. Meanwhile, C-UNet obviously got better segmentation results than the other four model, especially for small roads. Therefore, C-UNet obtained better results than other models and achieved state-of-the-art results.

## 5. Discussion

This section first identifies the simulation of our work and the organization of this paper, then discusses the open research lines generated by this work, establishing a roadmap for future works and improvements. These can be concluded in four aspects: simulation of our work, the structure of this paper, further research and application of our work.

### 5.1. Simulation of Our Work

In order to implement the model, we use Python and the open source neural network library Pytorch.

For the image processing, we first split a remote sensing image into 9 subimages, which saves memory space while training the model; then, we shuffle the images to finish the image processing.

For building the model C-UNet, we first employ the UNet model to get the first segmentation result. Secondly, we use the fixed threshold erasing method to erase the obvious segmentation regions. Thirdly, we construct the MD-UNet to do the second segmentation. Finally, we fusion the results of the first and the second segmentation.

For the loss function, we use the traditional binary cross-entropy loss, the road part in the segmentation results is 1, and the non-road part is 0. For the parameter part of C-UNet, we use the Adam optimizer to optimize, the learning rate is set to 0.0002, the epoch is set to 15, and the learning rate in the 9th stage becomes 10 times of the original. Our code runs on Tesla K40. In order to visualize the results (after the network was trained and tested), we use the matplotlib library. For the obtained segmentation feature map, we reset the image larger than 0.5 to 255, and we reset the other parts to 0 for visualization.

### 5.2. The Structure of this Paper

This paper includes six parts. They are Introduction, UNet, C-UNet, Experimental Results, Discussion, and Conclusions, respectively.

Specifically, in the Introduction section, we first explained the significance of road extraction and pointed out the challenge of road extraction. Then, we analyzed the methods proposed for road extraction from remote sensing images, summarized the existing problems, and presented our method.

Secondly, in the UNet section, we briefly introduced the overall architecture of UNet, and then gave the specific parameter values of UNet. This is the basis of our model.

Thirdly, in the C-UNet section, we first gave an overall depiction of C-UNet, including its flowchart, the model framework, and mathematical description. Later on, we described the four modules successively in detail. Finally, we introduced the loss function in the training process.

Fourthly, in the Experimental results section, we first described the public dataset used in our experiments. Then, we specified the implementation details and the evaluation indicators. Later on, we conducted a series of ablation studies to evaluate the effectiveness of each module of our model. Finally, we compared it with other four state-of-the-art methods on road segmentation from remote sensing images.

Fifthly, in the Discussion section, we explained the simulation of our model and point out the future research, as well as showed the applications of our model.

Finally, in the Conclusion section, we give a conclusion of our work, analyzed the reason of its better performance, and pointed out the future research direction.

### 5.3. Further Research

In the future, we will take C-UNet as a more general model and further improve its performance from two aspects. One is to combine with the attention scheme to select important features to segment the roads. Attention is the main means of the brain nervous system to solve the problem of information overload. It is a resource allocation scheme in the case of limited computing power, which allocates computing resources to more important information. With the help of attention scheme, we can guide C-UNet to pay more attention on roads, while ignoring other objects, such as buildings, rivers, cars, etc.

The other is to simplify its parameters with the light-weight methods, with the purpose of real-time segmentation. To improve the representation ability of C-UNet, we need to build a deep enough architecture and design a complex enough network structure. This means a long model training cycle and more machine memory. Therefore, it is a problem worthy of further study to build a lightweight model and to speed up the convergence of the model.

### 5.4. Application of C-UNet

This work has many applications. First, it is often difficult to acquire ground information in disaster relief if the ground object targets in the natural disasters (such as earthquakes, landslides, and torrential rain) are seriously damaged. This work can help to analyze the disaster situation quickly and conveniently. Secondly, roads play an important role in urban development, and this work can assist the government in road planning. Last but not least, considering the high similarity between roads and retinas, this work can be directly employed to extract retinas from medical images, helping doctors to better diagnose and treat diseases.

## 6. Conclusions

In this paper, we proposed a new model for road extraction in remote sensing images. Our proposed model includes four modules to extract the road information successively, which is demonstrated to be more suitable for extracting road information from high-resolution remote sensing images. With the help of complementary learning, a standard UNet is utilized to extract relatively obvious roads, and then the MD-UNet is introduced to extract complement and finer road information. Utilizing the multi-scale dilated convolution, MD-UNet can increase the receptive field without reducing the resolution of the feature map. This makes it better cope with different width, length, and shapes of roads. Ablation studies indicated that the proposed model and its submodules were effective. Comparative experiments showed that the proposed model outperformed the state-of-the-art methods.

In the future, we will extend the proposed C-UNet to segment different types of roads, such as forest and gravel roads. Besides, we will take it as a more general model to combine with attention-based methods and light-weight methods to make its performance better.

## Figures and Tables

**Figure 1 sensors-21-02153-f001:**
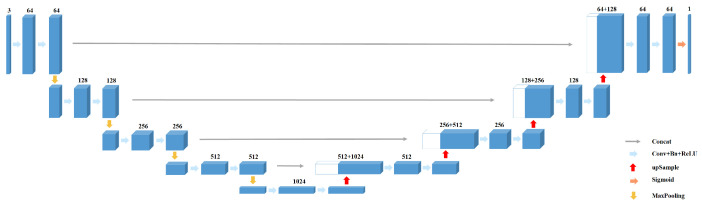
The architecture of UNet [38].

**Figure 2 sensors-21-02153-f002:**
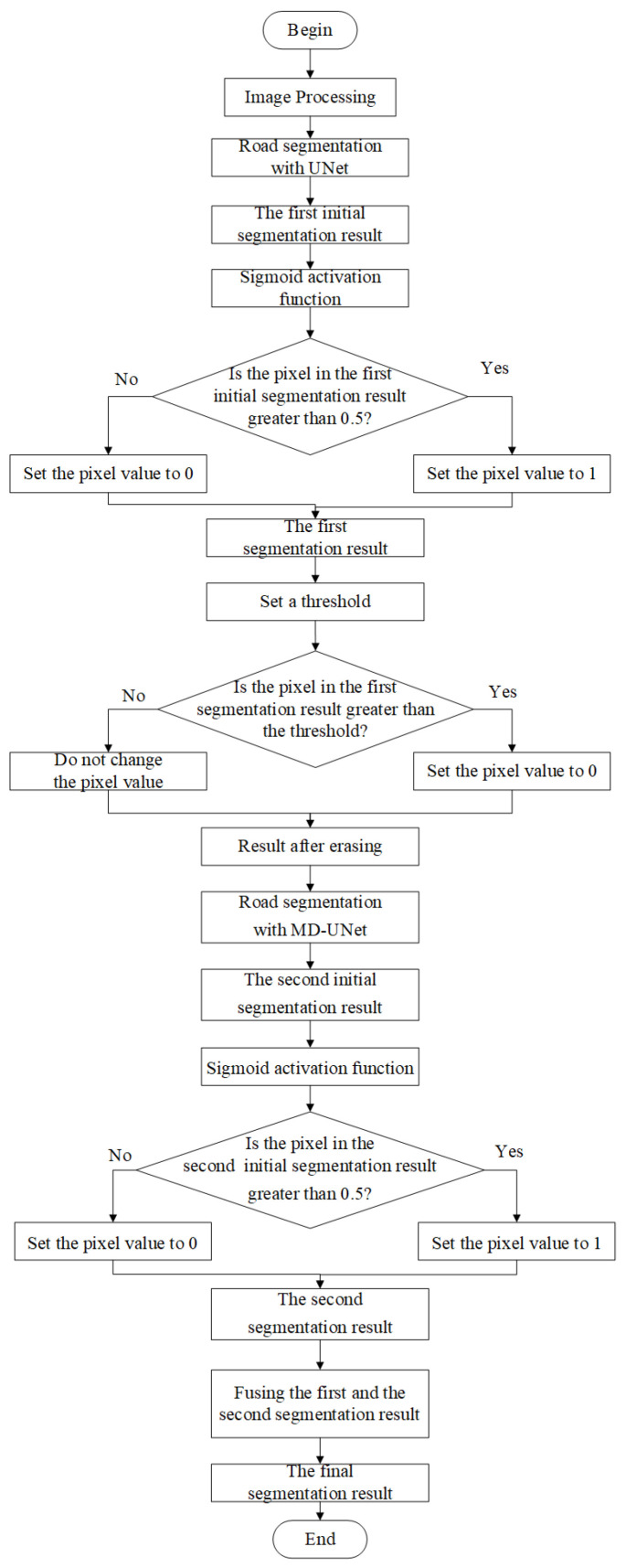
The flowchart of complement UNet (C-UNet).

**Figure 3 sensors-21-02153-f003:**
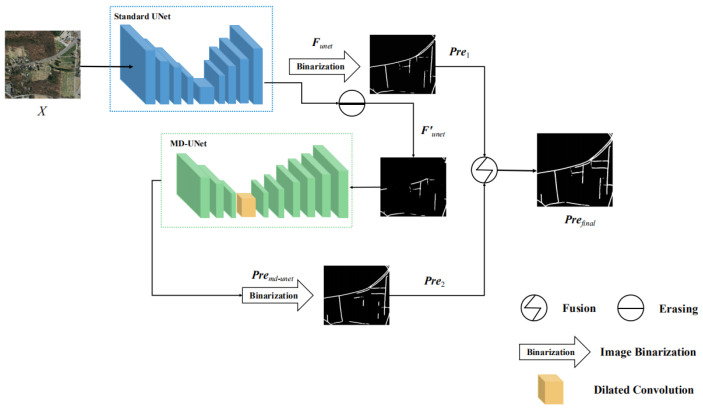
The architecture of C-UNet.

**Figure 4 sensors-21-02153-f004:**
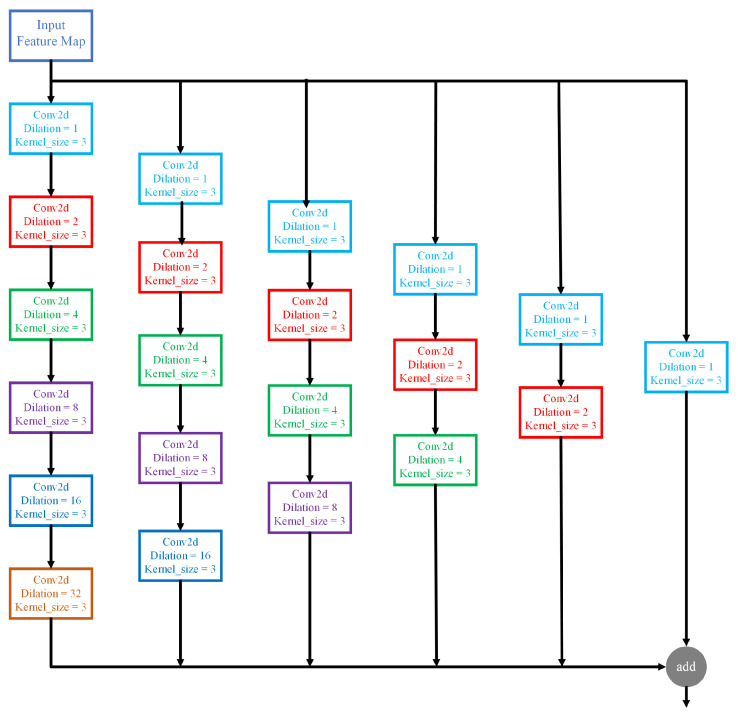
The module of multi-scale dilated convolution.

**Figure 5 sensors-21-02153-f005:**
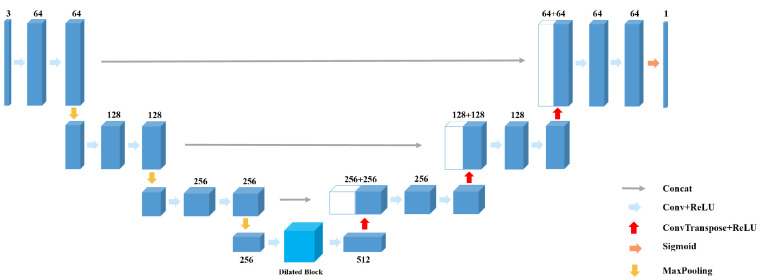
The architecture of multi-scale dilated convolution UNet.

**Figure 6 sensors-21-02153-f006:**
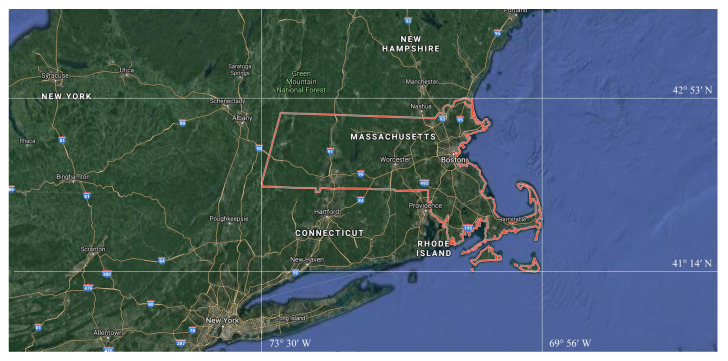
The geo-referenced map of Massachusetts.

**Figure 7 sensors-21-02153-f007:**
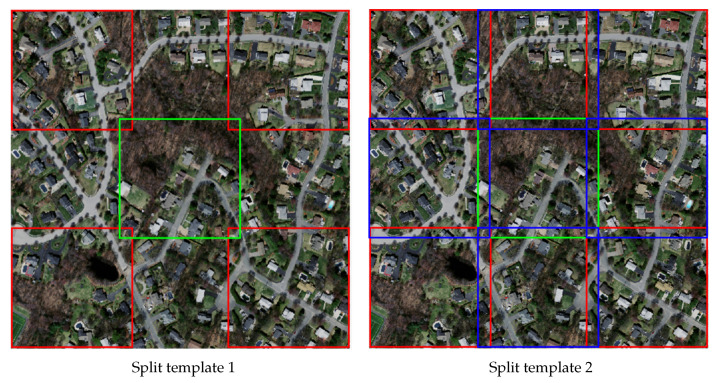
The dividing method. For the left image, we divide the image with five with five boxes, as denoted in the image with red boxes and the green box, getting 5 different sub-images. For the right image, we divide the image with another five boxes, as denoted in the image with red boxes, the blue boxes, and the green box, achieving another 5 sub-images.

**Figure 8 sensors-21-02153-f008:**
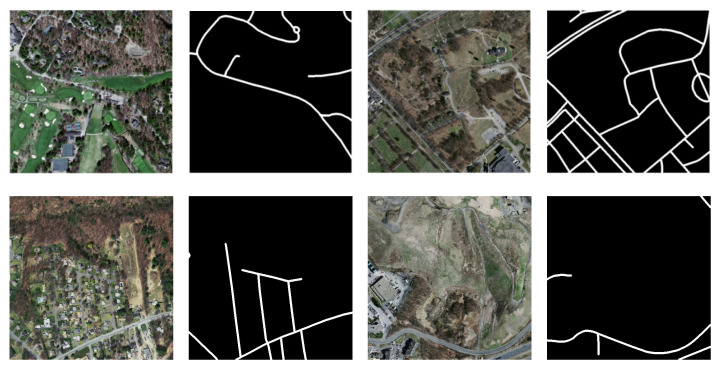
Examples of images in the Massachusetts dataset.

**Figure 9 sensors-21-02153-f009:**
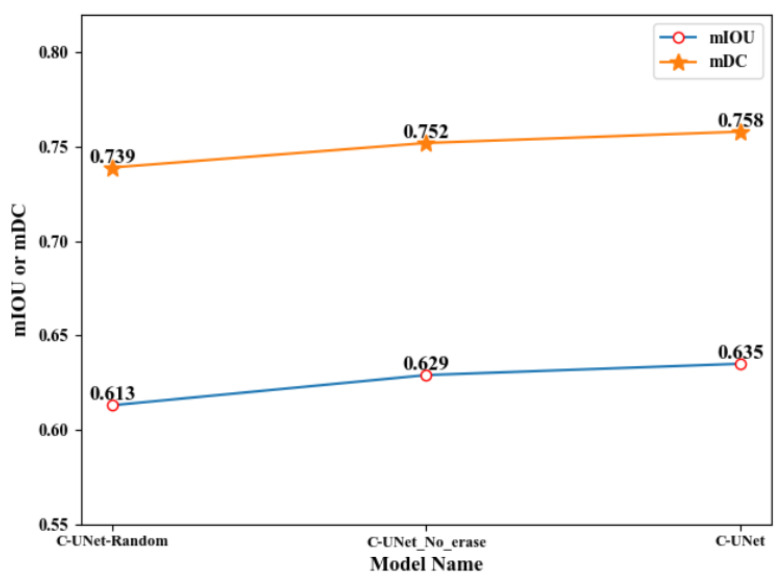
Line chart of C-UNet with different erasing methods.

**Figure 10 sensors-21-02153-f010:**
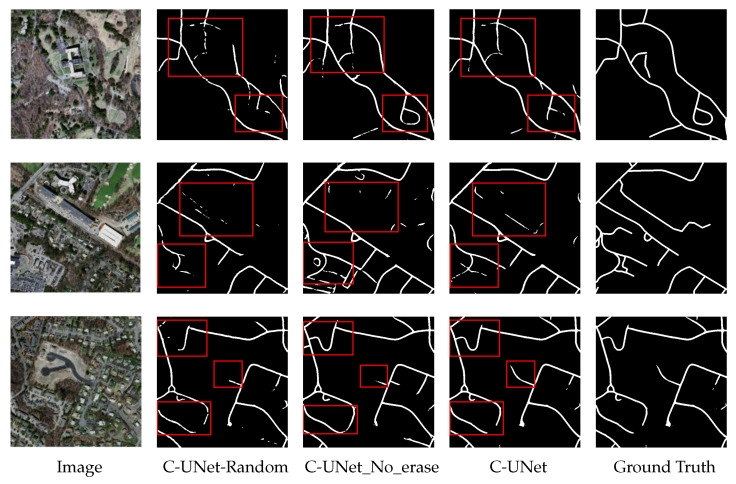
The segmentation results of three different methods.

**Figure 11 sensors-21-02153-f011:**
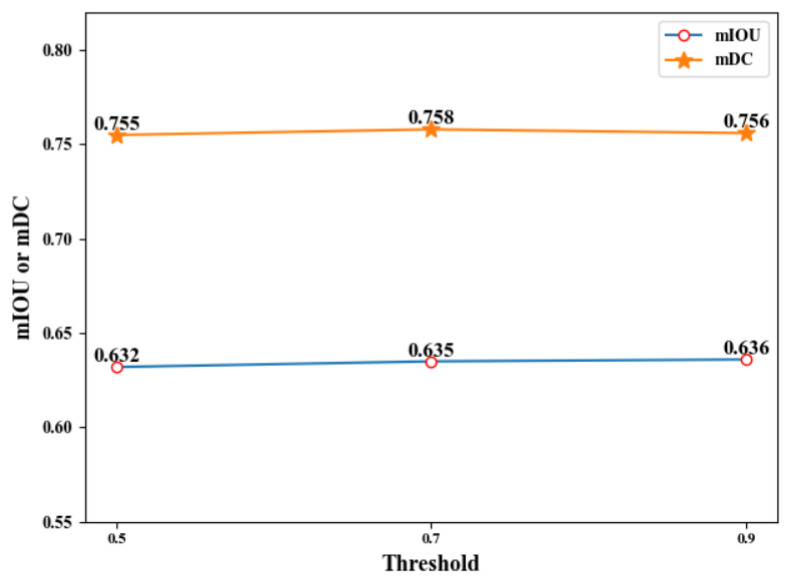
Line chart of C-UNet with different erasing thresholds.

**Figure 12 sensors-21-02153-f012:**
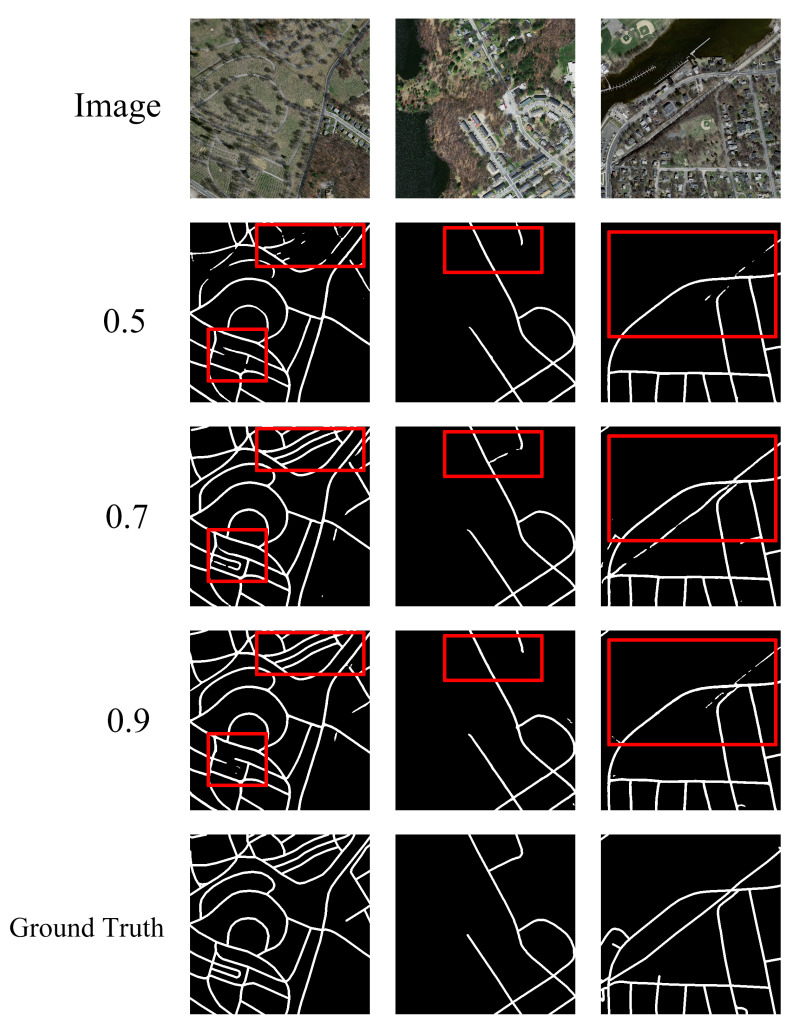
The final results of C-UNet with different erasing thresholds.

**Figure 13 sensors-21-02153-f013:**
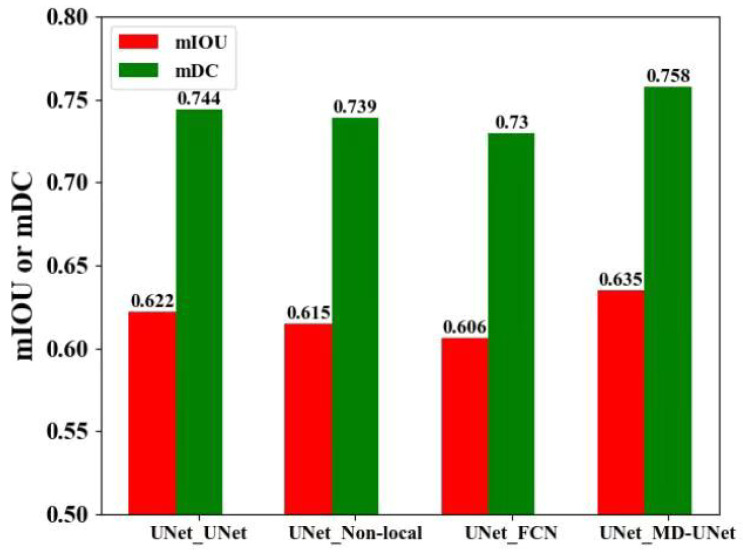
Bar graph of segmentation results of C-UNet with different models in the second stage.

**Figure 14 sensors-21-02153-f014:**
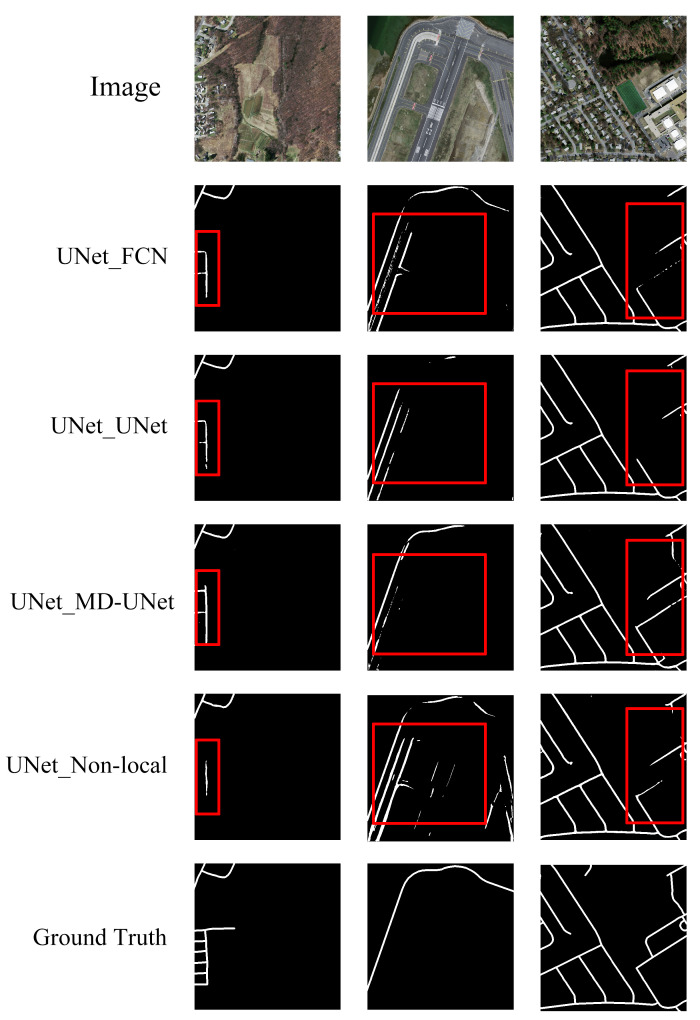
The final results of C-UNet with different models in the second stage.

**Figure 15 sensors-21-02153-f015:**
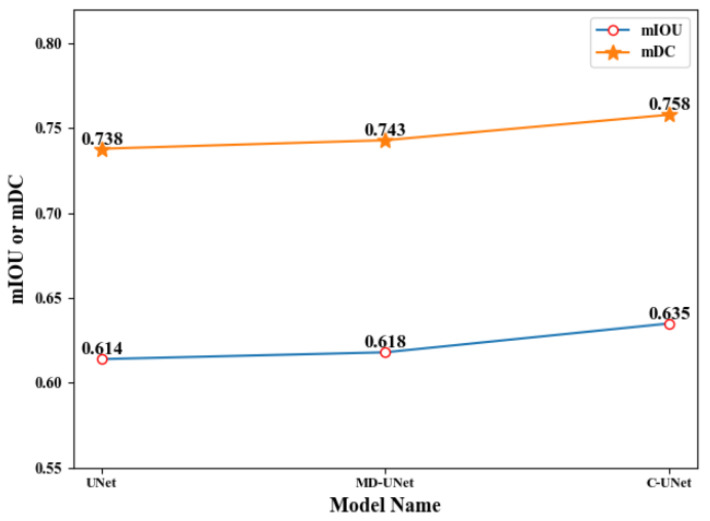
Line chart of results before and after fusion.

**Figure 16 sensors-21-02153-f016:**
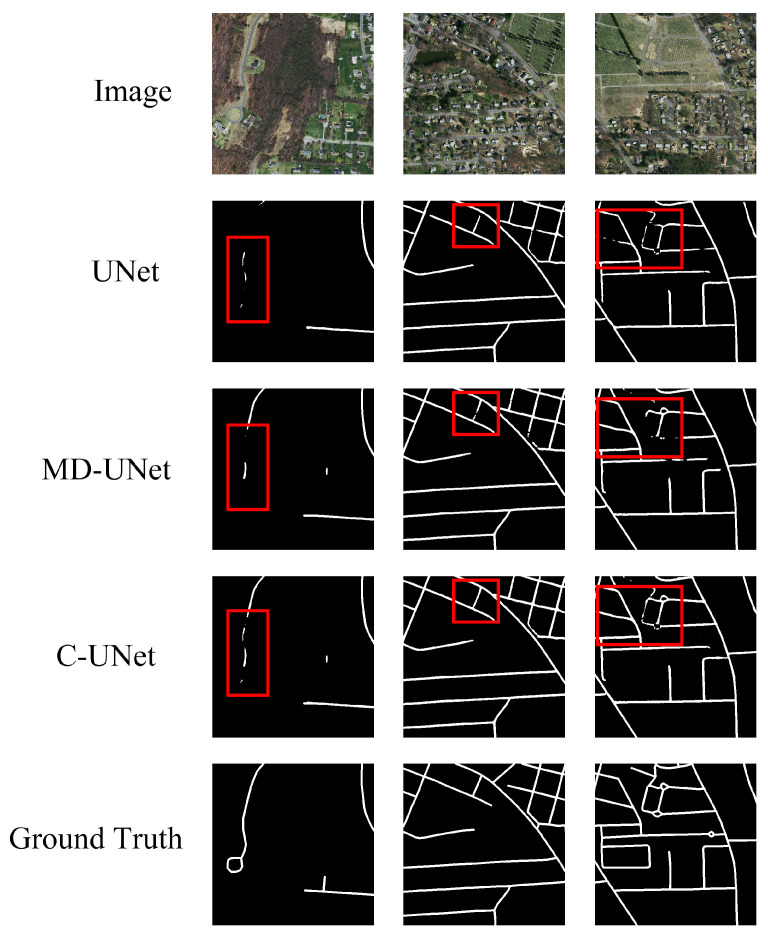
Image of results before and after fusion.

**Figure 17 sensors-21-02153-f017:**
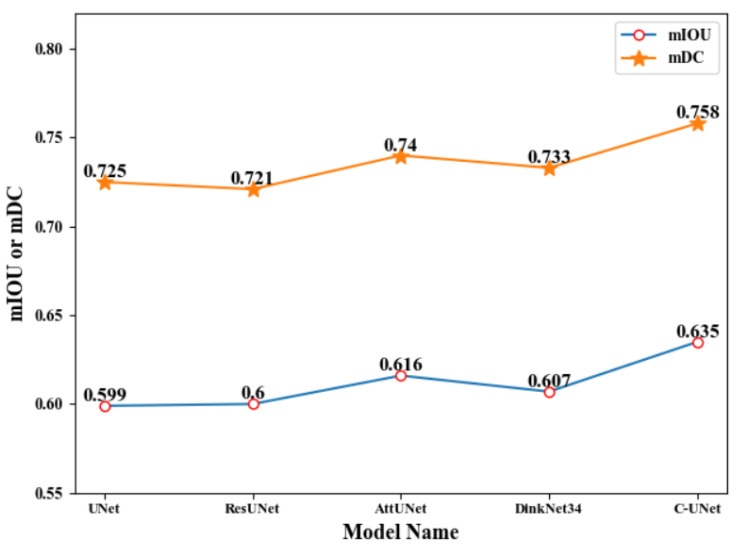
Line chart of different model performance.

**Figure 18 sensors-21-02153-f018:**
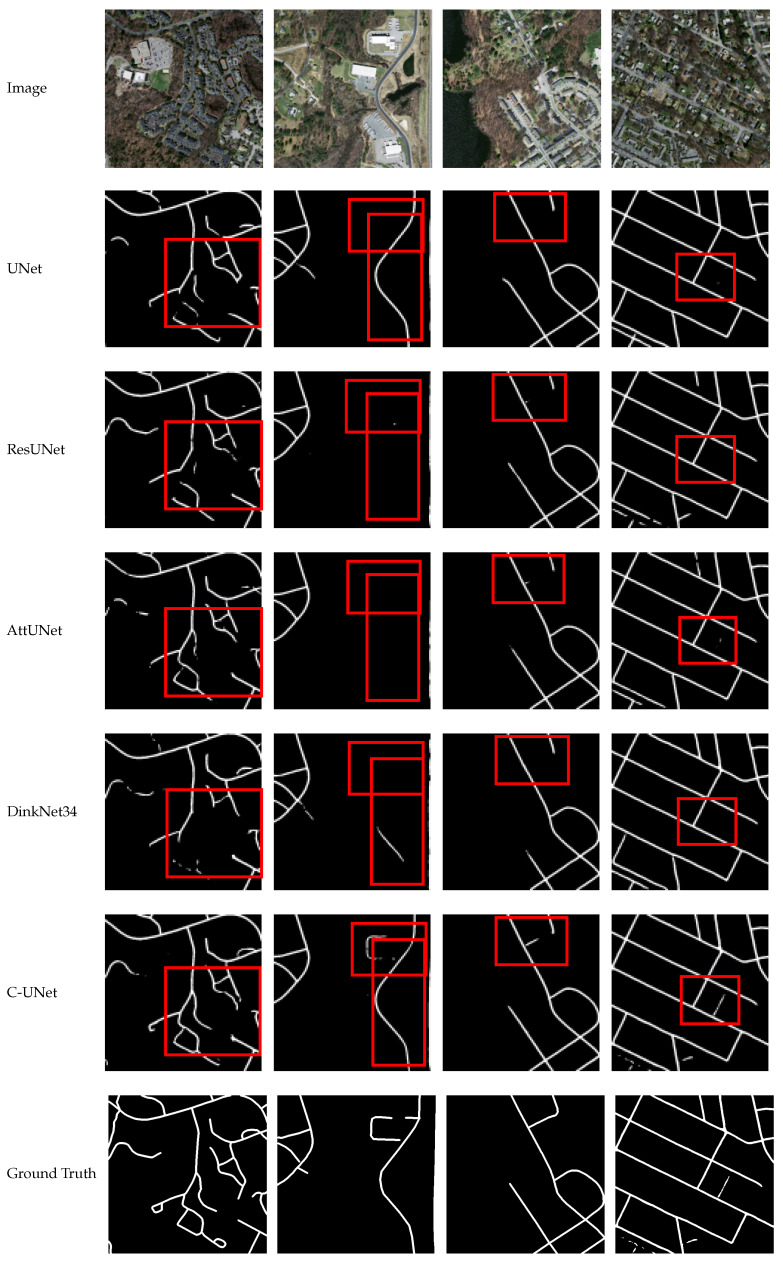
Segmentation results of different models.

**Table 1 sensors-21-02153-t001:** Results of different erasing methods.

Model	mIOU	mDC
C-UNet-Random	0.613	0.739
C-UNet_No_erase	0.629	0.752
**C-UNet**	**0.635**	**0.758**

**Table 2 sensors-21-02153-t002:** Performance of C-UNet with different erasing thresholds.

Threshold	mIOU	mDC
0.5	0.632	0.755
0.7	0.635	**0.758**
0.9	**0.636**	0.756

**Table 3 sensors-21-02153-t003:** Segmentation results of C-UNet with different models in the third module.

Model	mIOU	mDC
UNet_UNet	0.622	0.744
UNet_Non-local	0.615	0.739
UNet_FCN	0.606	0.730
**UNet_MD-UNet(C-UNet)**	**0.635**	**0.758**

**Table 4 sensors-21-02153-t004:** Comparison of results before and after fusion.

Model	mIOU	mDC
UNet	0.614	0.738
MD-UNet	0.618	0.743
C-UNet	**0.635**	**0.758**

**Table 5 sensors-21-02153-t005:** Performance of different models.

Model	mIOU	mDC
UNet	0.599	0.725
ResUNet	0.600	0.721
AttUNet	0.616	0.740
DinkNet34	0.607	0.733
**C-UNet**	**0.635**	**0.758**

## Data Availability

Not applicable.

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
