# Peer review of "C-UNet: Complement UNet for Remote Sensing Road Extraction"

_sensors, 2021, doi:10.3390/s21062153_

Round 1

Reviewer 1 Report

Dear authors, the manuscript is detailed and informative but it has some inadequacies. I shall highlight them and the authors might improve the quality and readability of this research paper accordingly.

1. Please include a geo-referenced map of the study area shown within the World map. Please include a high-resolution figure/image.

2. All the original image segments, including the ground truth segments should be geo-referenced.

3. The methods are not correctly described and sufficiently informative to allow the replication of the research. The methodology of this research work should be described by a schematic and conceptual flowchart.

4. The most relevant data-results should be summarized and demonstrated by a graph and a corresponding table.

5. Please, highlight the outliers in the table/s and graphs, where relevant.

6. All the given table/s in the manuscript should be graphically represented and validated. The graphs could be combined where appropriate.

7. The techniques and/or models presented and mentioned in the manuscript require sufficient details (including calibration, sensitivity analysis and validation) to allow other researchers to develop and test the applications later on. Please include the parameters that I have mentioned here.

8. Please, briefly add future perspectives and further applied applications of this specific research work in the discussion section, before the conclusion.

9. Only a few references have been included in the manuscript. Please add relevant references to enhance the global importance of the paper.

10. The English language of the paper needs a thorough revision. Please have the manuscript proofread by competent authorities. Please make it coherent dealing with each section at a time and linking one paragraph to the next having focus, scale, novelty, importance, clarity and rigour.

11. Please include all the Software codes that have been implemented in this particular research work in the appendix section for independent simulation, testing, validation and integration.

Author Response

lease see the attachment

Reviewer 2 Report

The paper presents a road extraction method for remote sensing images with a complement UNet (C-UNet). The introduction focus on the different technologies for getting remote sensing images. I would suggest to add a sentence of the aerial infrared thermography applied to the road analysis. For more information refer to a general review on the topic https://doi.org/10.1016/j.rser.2017.10.031. The structure of the paper is not inserted. Please add a part with the discussion of the sims and the structure of the paper. Also the difference between u net and c net is not clear. Please add a introduttive sentence that explicate the different approach and application of these technologies. The comprehension of the results is very difficult without the description and f the structure of the paper. Also, is important to show better the input data. In the conclusion show better using bullet points the most important outcomes of the research and its novelty compared to other research on the same topic.

Reviewer 3 Report

1) How is the binary operation developed? Is it Otsu method, please discuss.

2) what is the main novelty of proposed approach? is the developed method is generic and can employed for other application? why do you compare with convolutional networks (for biomedical image segmentation)? 

3) The introduction/review part is too shallow. More articles should be analysed, e.g. segmentation based on graph cuts method and Markov random fields method is developed in [1] and applied for road defects detection and classification.

4) The conclusion is again pretty shallow: just 6 lines. Pls include more analysis: why are you better comparing with competing methods? is it possible to extend your approach on forest/gravel roads?  

[1] H. T. Nguyen, L. T. Nguyen et al.  “A robust approach for road pavement defects detection and classification”, J. Comp. Eng. Math., 3:3 (2016), 40–52, DOI: 10.14529/jcem160305

Round 2

Reviewer 1 Report

Dear authors, I am okay with the changes made. Thanks.

Reviewer 2 Report

You improved a lot your paper 

Reviewer 3 Report

the revised version can be accepted.